# THE LAST TOKEN IS ENOUGH: LIGHTWEIGHT CONTRASTIVE DECODING FOR MITIGATING HALLUCINATIONS IN LARGE VISION-LANGUAGE MODELS

## ABSTRACT

Contrastive decoding is a mainstream approach for mitigating hallucinations, which actively induces hallucinations and compares them with the original output to calibrate the logits of the next token, thereby enhancing the response reliability of Large Vision-Language Models (LVLMs). Although contrastive decoding has shown promising effectiveness in suppressing hallucinations, the process of inducing hallucinations inevitably incurs additional computational overhead and considerable inference latency, an issue largely overlooked in existing studies. To this end, we present LightCD, a lightweight and efficient contrastive decoding method that achieves better hallucination mitigation while maintaining high computational efficiency. Specifically, we observe that LVLMs tend to aggregate learned information into a small number of tokens, which play a critical role in supporting the final output. The last token, used for next-token prediction, leverages the attention mechanism to "see" and summarize this information in order to produce the final result. In light of this, we revisit the process of inducing hallucinations in the current contrastive decoding paradigm and propose processing solely on the last token in the final layer of LVLMs to replace full-sequence inference. LightCD introduces two key modules: Selective Attention Perturbation, which identifies and filters critical information from attention heads containing factual evidence; and a Residual-free Mechanism, which suppresses cross-layer information propagation to generate desired hallucinations. Finally, the model completes contrastive decoding by subtracting the hallucination logits from the original token logits. Extensive experiments demonstrate that LightCD significantly outperforms existing methods in mitigating hallucinations while achieving a 2× increase in inference speed. [1]

## 1 INTRODUCTION

The emergence of Large Vision-Language Models (LVLMs) (Liu et al., 2023; 2024b; Li et al., 2023a; Dai et al., 2023; Bai et al., 2025) built upon Large Language Models (LLMs) (DeepSeek-AI et al., 2025; 2024; Yang et al., 2024; Chiang et al., 2023) has further enhanced our understanding and modeling of the real world, demonstrating impressive performance on complex multimodal tasks such as image understanding and visual question answering. However, the inherent hallucination problem in LVLMs poses significant challenges to their reliability and safety in real-world applications. Hallucination refers to cases where the model produces outputs that appear plausible yet contradict the factual content of the input image, such as incorrectly describing object existence or spatial relationships (Leng et al., 2024; Liu et al., 2024a; Chen et al., 2024a). Addressing hallucinations is therefore essential for the safe and trustworthy deployment of LVLMs in safety-critical domains.

Contrastive decoding (Li et al., 2023b) has recently emerged as a prominent strategy for mitigating hallucinations. This approach works by actively inducing potential hallucinations and, during decoding, correcting the next-token logits by subtracting the hallucinated outputs from the original predictions (Leng et al., 2024; Suo et al., 2025; Huang et al., 2024). Compared with supervised

---

[1]Our code is available at https://anonymous.4open.science/r/LightCD-3558

fine-tuning (Liu et al., 2024b; Zhu et al., 2025), contrastive decoding offers superior generalization and scalability. As a representative work, VCD (Leng et al., 2024) introduces Gaussian noise into visual inputs to amplify language bias and thus induce hallucinations. SID (Huo et al., 2025) leverages the introspective capability of LVLMs by preserving prompt-irrelevant visual regions to elicit hallucinations. Although these approaches have shown effectiveness in mitigating hallucinations, they share an inherent limitation—**low inference efficiency.**

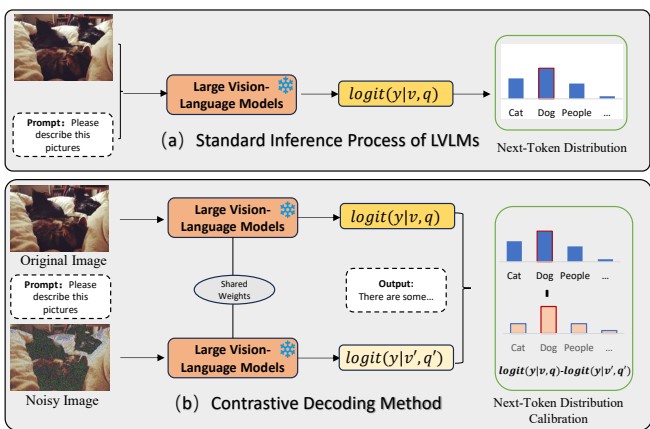

Figure 1: Standard Inference VS Contrastive Decoding (VCD as Example).

As illustrated in Figure 1, the hallucination-inducing stage in contrastive decoding requires an additional full forward pass compared with standard inference. This extra computation introduces substantial latency, which severely limits its applicability in real-time scenarios such as autonomous driving. However, this critical issue has been largely overlooked in existing studies. A critical yet unexplored question arises: **How can we leverage the benefits of contrastive decoding for hallucination mitigation while preserving inference efficiency?**

The inference latency of contrastive decoding primarily stems from the hallucination-inducing stage, and whether this stage truly requires the full computational cost is key to addressing the latency issue. Through empirical analysis, we observe that the attention distribution in the decision layer of LVLMs is highly skewed: The last token, used for next-token prediction, allocates the vast majority of its attention to a very small subset of tokens (see Section 3.3). In other words, these few tokens dominate the generation process Chen et al. (2024b). This insight provides a clear answer on whether full computation is necessary: **No, the last token is enough.**

Building on these observations, we propose LightCD, a lightweight and efficient contrastive decoding framework designed to mitigate hallucinations in LVLMs while preserving inference speed effectively. LightCD comprises two core components: **Selective Attention Perturbation (SAP)** and a **Residual-free mechanism**. Unlike methods that perturb from the input layer, LightCD induces hallucinations by intervening solely in the last token of the final LVLM layer, thereby substantially reducing computational overhead. Specifically, SAP identifies critical attention heads based on entropy and selectively filters the essential information they contain to elicit model hallucinations. The Residual-free mechanism then blocks information from previous layers, further amplifying the effect of SAP and producing high-quality hallucinatory outputs. Finally, contrastive decoding is applied to generate reliable responses. By optimizing the hallucination generation process, LightCD achieves a balance between inference efficiency and decoding effectiveness, delivering superior hallucination mitigation while achieving twice the inference speed of existing methods (Leng et al., 2024; An et al., 2025; Zhao et al., 2025). Our contributions can be summarized as:

- We highlight the problem of low inference efficiency in contrastive decoding for LVLMs—a limitation that constrains its applicability in real-world scenarios and has often been overlooked in prior work.

- We observed that the last token plays a key role in summarizing knowledge and revisited the construction of hallucinations in contrastive decoding, which inspired the design of LightCD. By optimizing hallucination generation, LightCD reduces inference cost by nearly half, effectively alleviating the latency in contrastive decoding.

- Extensive experiments on multiple LVLMs, including LLaVA1.5, InstrucBlip, and Qwen-VL, demonstrate the superior performance of LightCD in mitigating model hallucinations. These results also verify the generalizability of our method, as it consistently delivers strong performance across various types of LVLMs.

## 2 RELATED WORK

**Contrastive Decoding for Mitigating Hallucinations in LVLMs.** Recent studies (Leng et al., 2024; Chen et al., 2024a; Zhao et al., 2025; Yin et al., 2025) suggest that over-reliance on the language priors of LLMs is a major contributor to hallucinations in LVLMs. To address this issue, contrastive decoding (Li et al., 2023b; Leng et al., 2024) has been proposed to amplify halluci-nations through perturbations intentionally and subsequently calibrate the next-token probability distribution by subtracting the hallucinated output from the original output, thereby improving the reliability of generated content. The key challenge in contrastive decoding lies in how to generate high-quality hallucinations to eliminate language priors (Zhao et al., 2025). As an early approach, VCD (Leng et al., 2024) perturbs the image by introducing Gaussian noise to obtain hallucinated content. SID (Huo et al., 2025) uses the self-reflective capacity of LVLM by retaining less impor-tant tokens in the image input to amplify hallucinations. In addition to these, several methods (Cho et al., 2025; An et al., 2025) inspired by contrastive decoding have been proposed to enhance output reliability by adjusting the next-token logits. For example, ED (Cho et al., 2025) segments the input image into sub-images and uses attention maps to assign weights when combining their respective logits. While these approaches help suppress hallucinations, they also increase inference cost and la-tency—an issue often overlooked. To address this issue, we revisit the existing contrastive decoding paradigm with the aim of developing a lightweight approach that effectively mitigates hallucinations while maintaining computational efficiency.

## 3 PRELIMINARIES AND MOTIVATION

### 3.1 LVLMs INFERENCE PIPELINE

LVLMs align images to the semantic space of the LLM using an image encoder followed by a fully connected layer, producing a sequence of visual tokens $V = \{v_1, v_2, ..., v_m\}$, where $m$ is the number of visual tokens. Similarly, a tokenizer processes the input prompt into a sequence of text tokens $T = \{t_1, t_2, ..., t_n\}$, where $n$ is the length of the text token. These tokens are then concatenated to form the input sequence $I = \{V, T\}$, which is fed into the LLM. The input $I$ undergoes processing through $L$ layers of Transformer blocks, each consisting of multi-head attention layers and feed-forward networks:

$$\hat{\boldsymbol{I}}^{(l)} = \boldsymbol{I}^{(l-1)} + \sum_{n=1}^{H} Attn_n^{(l)}(\boldsymbol{I}^{(l-1)}) \tag{1}$$

$$\boldsymbol{I}^{(l)} = \hat{\boldsymbol{I}}^{(l)} + \boldsymbol{FFN}^{(l)}(\hat{\boldsymbol{I}}^{(l)}) \tag{2}$$

Where $Attn_n^{(l)}$ denotes the attention operation of the $n$-th head in the $l$-th layer, $FFN^{(l)}$ denotes the feed-forward neural network in the $l$-th layer, and $H$ denotes the number of attention heads. After the final layer $\boldsymbol{L}$, the LVLM generates a probability distribution over the vocabulary, from which the next token $y_t$ is autoregressively sampled:

$$y_t \sim p(y_t \mid V, T, y_{<t}) = Softmax(fc(\boldsymbol{I}^{(L)})) \tag{3}$$

Here, $y_t$ and $y_{<t}$ denote the token generated at step $t$ and the sequence of tokens generated before step $t$, respectively. The $fc(\cdot)$ denotes a projection layer that maps the hidden representation $\boldsymbol{I}^{(L)}$ to a probability distribution over a vocabulary of size $M$.

### 3.2 CONTRASTIVE DECODING PARADIGM IN LVLMs

Contrastive decoding calibrates the model's output by subtracting the logits of hallucinated from those of the original output, thereby guiding LVLMs to generate more reliable responses. Taking

VCD (Leng et al., 2024) as an example, it introduces hallucinated content by adding Gaussian noise to the original image, which can be formally expressed as follows:

$$p_{vcd}(y_t|V, V^d, T, y_{<t}) = Softmax[(1 + \alpha)logit_\theta(y_t|V, T, y_{<t})$$
$$- \alpha \cdot logit_\theta(y_t|V^d, T, y_{<t})] \tag{4}$$

Where $V^d$ denotes the perturbed image. Current research on contrastive decoding focuses mainly on constructing hallucinated inputs for decoding. However, the generation of hallucinations doubles the inference cost, resulting in significant inference latency.

## 3.3 SHOULD HALLUCINATION GENERATION PAY THE FULL COMPUTATIONAL PRICE

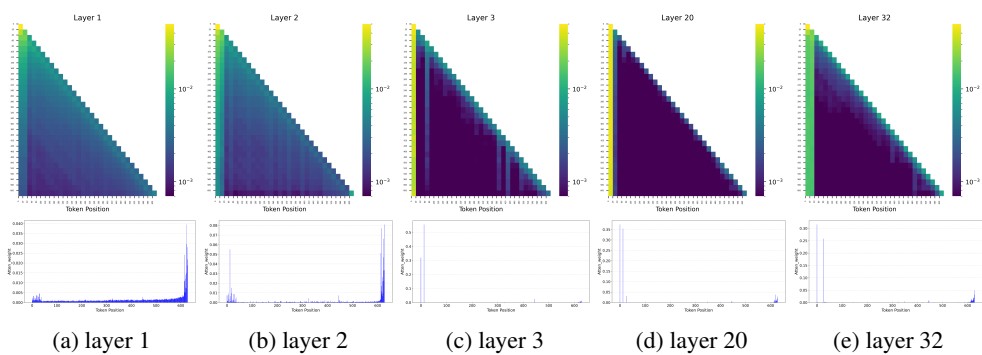

(a) layer 1     (b) layer 2     (c) layer 3     (d) layer 20     (e) layer 32

Figure 2: Visualizing attention weights across different layers of LLaVA 1.5-7B. (Top) Full attention heatmap across all tokens. (Bottom) Attention map of the last token, where the horizontal axis denotes the token position index. The model first identifies anchor tokens in the early layers, and as the depth increases, it allocates progressively more attention to these anchors.

Although contrastive decoding effectively mitigates hallucinations, it also doubles inference time, as LVLMs must perform an additional forward pass for hallucination generation. This raises a natural question: Is it truly necessary for hallucination generation to incur the full computational cost of standard inference?

To answer this question, we first need to clarify the complete inference process of an LVLM. As described in Section 3.1, the image and the prompt are transformed into a sequence of tokens, which are then fed into the LVLM. These tokens are processed through multiple Transformer layers to predict the next token, and the LVLM performs this procedure autoregressively until a full response is generated. Intuitively, the prediction of the next token depends only on the representation of the last token in the final layer. To further investigate this process, we visualize the attention patterns of several intermediate layers, as illustrated in Figure 2. It shows that the distribution of attention is highly uneven. Starting from the third layer, the LVLM concentrates most of its attention on only a small subset of tokens, which serve as "anchors" (Chen et al., 2024b). As the computation proceeds, the model progressively accumulates the knowledge learned at each layer onto these anchor tokens. In the final layer, the last token in the sequence relies heavily on these anchors to integrate information, thereby producing the prediction for the next token. Statistical analysis shows that in LLaVA, roughly 2% of the tokens account for more than 80% of the last token's attention, with an even more pronounced concentration in InstructBLIP, exhibiting a long-tail distribution. Furthermore, owing to the autoregressive nature of the model, these anchor tokens—apart from a few sink tokens (Kang et al., 2025)—are predominantly distributed toward the right end of the sequence. We attribute this phenomenon to the fact that the input image and prompt are segmented into a large number of tokens, and attending to too many tokens simultaneously would dilute the information. As a result, the model identifies a set of anchor tokens in the early layers and progressively accumulates knowledge onto them in subsequent layers. When it ultimately needs to predict the next token, the last token primarily attends to the knowledge encoded in these anchors. In other words, these anchor tokens dominate the prediction of the next token.

This observation inspires the design of **LightCD**, an efficient and plug-and-play contrastive decoding method. LightCD induces and calibrates hallucinations by intervening solely on the **last token**,

significantly reducing inference overhead and establishing a new decoding paradigm: **The Last Token is Enough**.

# 4 METHOD: THE LAST TOKEN IS ENOUGH

In this work, we propose LightCD(Figure 3), a lightweight contrastive decoding framework that mitigates hallucinations while maintaining efficient inference in LVLMs. LightCD comprises two core components: Selective Attention Perturbation and a Residual-free mechanism. In this section, we provide a detailed description of the LightCD architecture.

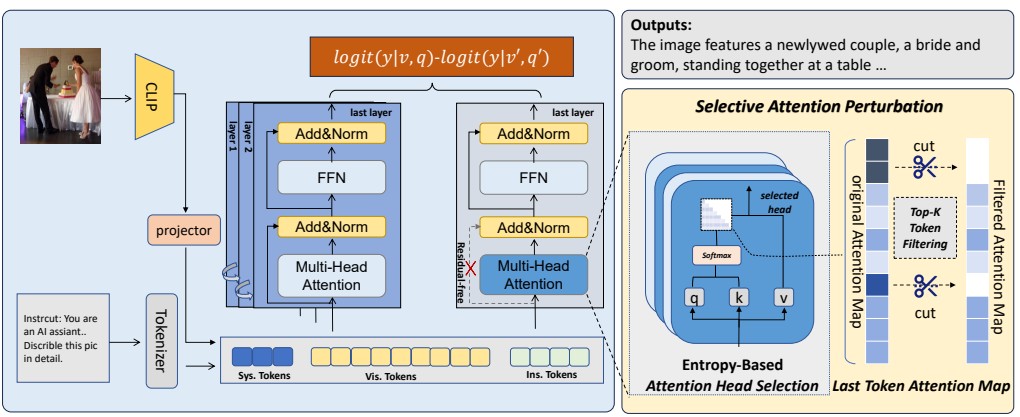

Figure 3: Overview of LightCD. LightCD employs Selective Attention Perturbation(SAP) and a Residual-free mechanism to process only the last token in the final layer of LVLMs, enabling lightweight contrastive decoding. The SAP identifies attention heads and filters critical information, while the Residual-free mechanism disrupts information propagation to amplify hallucinations.

## 4.1 SELECTIVE ATTENTION PERTURBATION

The hallucination generation stage is the primary source of inference latency in contrastive decoding. Taking VCD as an example, hallucinations are produced by injecting Gaussian noise into the image and performing an additional full forward pass to obtain the contrastive signal. This process incurs computational overhead equivalent to standard inference. Moreover, the use of random Gaussian noise may introduce unintended hallucinations, leading to unstable decoding (Huo et al., 2025).

The visualization analysis reveals that the internal attention distribution of LVLMs is highly imbalanced (Section 3.3). Only an extremely small subset of tokens provides the critical evidence for the last-token prediction, effectively dominating the next-token generation process. By intervening in the attention of the last token at the decision layer and suppressing these critical tokens, hallucinations can be reliably induced without propagating perturbations from the input layer. This design not only avoids the full computational cost associated with input-level perturbations but also eliminates the instability caused by random noise.

Motivated by these observations, we introduce Selective Attention Perturbation (SAP), which operates by intervening exclusively in the multi-head attention computation of the last token in the final layer. SAP suppresses the key anchor tokens that dominate next-token prediction, thereby weakening their semantic contribution and inducing hallucinations in a controlled and targeted manner. Through targeted suppression of anchor tokens, SAP produces consistent and semantically meaningful hallucination, which are well-suited for subsequent contrastive decoding.

SAP operates in two steps: (1) Attention Head Selection, which identifies attention heads that most sensitive to semantic evidence; and (2) Top-$\gamma$ Token Filtering, which suppresses the most dominant tokens within those heads.

**Attention Head Selection.** In multi-head attention mechanisms, different heads typically capture distinct types of information, existing methods often train an auxiliary network to identify target

heads that are highly relevant to the model's output(Chen et al., 2024a). However, such methods introduce additional computational overhead. To maintain efficiency, LightCD proposes an entropy-based attention head selection strategy. The underlying intuition is that a lower entropy indicates a more concentrated attention distribution, suggesting that the head is more "focused" and thus more likely to capture critical information. Consequently, such heads are identified as the targets for perturbation.

Specifically, we first compute the attention weights $\mathbf{A} \in \mathbb{R}^{H \times 1 \times T}$ of the last token over the entire sequence, where $H$ and $T$ denote the number of attention heads and sequence length, respectively. And then calculate the attention entropy for each head.

$$\mathbf{A} = Softmax\left(\frac{qK^\top}{\sqrt{d_k}}\right), \quad p^{(h)} = \mathbf{A}^{(h)}[-1]$$
$$s_h = -\sum_{j=1}^{T} p_j^{(h)} \log(p_j^{(h)} + \epsilon)$$

(5)

Here, $p^{(h)} \in \mathbb{R}^T$ denotes the attention distribution of the last token in the $h$-th head, $\epsilon$ is the smoothing factor. Finally, the top $k$ heads with the lowest entropy are selected as the target heads.

$$\mathcal{H}_{\text{target}} = \text{TopK}_k\left(\{s_h\}_{h=1}^{H}\right)$$

(6)

The target attention head set is denoted as $\mathcal{H}_{\text{target}} \subseteq \{1, 2, \ldots, H\}$, where H denotes the total number of attention heads.

**Top-$\gamma$ Token Filtering.** After identifying the target heads, we selectively forget the key information within these heads by modifying their attention matrices — specifically, setting the Top-$\gamma$ weights in target heads to zero. The updated attention is then combined with the value matrix $V$ to produce an output that amplifies the hallucinated content.

$$\hat{A}^{(h)}[i] = \begin{cases} 0, & \text{if } i \in \text{Top-}\gamma(A^{(h)}[i]) \text{ and } h \in \mathcal{H}_{\text{target}} \\ A^{(h)}[i], & \text{otherwise} \end{cases}$$

(7)

$$O^{(h)} = \hat{A}^{(h)} V^{(h)}, O = Concat(O^{(1)}, O^{(2)}, ..., O^{(H)})W^o$$

(8)

where $W^o$ denotes the output projection matrix.

## 4.2 RESIDUAL-FREE MECHANISM

In LVLMs, residual connections serve as a crucial mechanism for sustaining information flow and preventing degradation as network depth increases. By directly adding a layer's input to its output(Eq. 1 and Eq. 2) to enable cross-layer information propagation, thereby facilitate more stable and effective representation learning. However, during hallucination generation, cross-layer information propagation can restore the critical knowledge filtered out by SAP, thereby compromising the quality of the induced hallucinations—an outcome that is undesirable. To address this issue, we remove the residual connection in the attention mechanism during the hallucination generation process and accordingly modify Eq. 1 as follows:

$$\hat{\boldsymbol{I}}^{(l)} = \sum_{n=1}^{H} Attn_n^{(l)}(\boldsymbol{I}^{(l-1)})$$

(9)

By removing the residual mechanism during hallucination generation, we break the cross-layer information propagation in the final layer, thereby suppressing knowledge learned in earlier layers and inducing the desired hallucinations for effective contrastive decoding.

## 5 EXPERIMENTS AND ANALYSIS

### 5.1 EXPERIMENTAL SETTING

**Datasets and Baselines.** To comprehensively evaluate the effectiveness of our proposed LightCD, we conduct experiments on three representative LVLMs—LLaVA (Liu et al., 2024b), Instruct-BLIP (Dai et al., 2023), and Qwen-VL Bai et al. (2023), across five benchmark datasets. (1)

**POPE** (Li et al., 2023c) is designed to evaluates object hallucination by requiring yes/no judgments about the presence of objects.(2) **MME-Hallucination** (Fu et al., 2025) is a comprehensive benchmark for object hallucination in LVLMs, covering object-level (existence and counting) and attribute-level hallucination (color and spatial location).(3) **CHAIR** Rohrbach et al. (2018) evaluates object hallucination by examining inconsistencies between generated image captions and the visual content.(4) **MMBench** (Liu et al., 2024c) assesses a model's multimodal understanding across 20 dimensions. (5) **MM-Vet** Yu et al. (2024) evaluates LVLM capabilities in six aspects, including recognition, OCR, knowledge, language generation, spatial awareness, and math—using GPT-based scoring.

The baselines used in our experiments include VCD (Leng et al., 2024), ICD (Wang et al., 2024), M3ID (Favero et al., 2024), Woodpecker (Yin et al., 2024), HALC Chen et al. (2024c), DoLa Chuang et al. (2024), OPERA Huang et al. (2024), SID (Huo et al., 2025) and ONLY (Wan et al., 2025). In all experiments, we adopt a sampling-based decoding strategy, where the next token is directly sampled from the post-softmax probability distribution by default.

**Experimental Details.** For fair comparison, we follow the settings used in previous works (Leng et al., 2024), In all experiments, the hyperparameter $\alpha$ in Equation 4 is set to 1. For the additional hyperparameters introduced by LightCD, we set $k = 28$ in Equation 6 and $\gamma = 0.3$ in Equation 7 for LLaVA and InstructBLIP. For Qwen-VL, we use $k = 20$ and $\gamma = 0.3$. In addition, we compute entropy (Eq. 5) using only the ten largest attention weights rather than the full sequence, so as to reduce the influence of numerous small values in the long-tail distribution. All experiments are conducted on an NVIDIA A100 GPU.

## 5.2 MAIN RESULTS

| Model | Method | Adversarial | | Popular | | Random | | AVG | |
|---|---|---|---|---|---|---|---|---|---|
| | | Accuracy | F1 Score | Accuracy | F1 Score | Accuracy | F1 Score | Accuracy | F1 Score |
| LLaVA-1.5 | Regular | 78.96 | 77.57 | 81.88 | 80.06 | 83.29 | 81.33 | 81.38 | 79.65 |
| | VCD | 80.88 | 81.33 | 85.38 | 85.06 | 87.73 | 87.16 | 84.66 | 84.52 |
| | M3ID | 78.23 | 80.22 | 84.30 | 84.95 | 87.50 | 87.67 | 83.34 | 84.28 |
| | SID | 80.26 | 81.57 | 85.37 | 85.16 | 88.16 | 87.97 | 84.60 | 84.90 |
| | LightCD(ours) | **84.56** | **84.42** | **88.00** | **87.46** | **89.90** | **89.69** | **87.49** | **87.19** |
| InstructBlip | Regular | 75.84 | 76.59 | 78.22 | 78.36 | 80.71 | 80.41 | 78.26 | 78.45 |
| | VCD | 79.56 | 79.52 | 81.47 | 81.07 | 84.53 | 83.68 | 81.85 | 81.42 |
| | M3ID | 77.47 | 79.14 | 80.97 | 81.85 | 86.67 | 86.41 | 81.70 | 82.47 |
| | SID | 79.80 | 80.73 | 81.40 | 81.94 | 87.83 | 87.34 | 83.01 | 83.34 |
| | LightCD(ours) | **81.90** | **81.93** | **83.50** | **82.70** | **88.76** | **88.22** | **84.72** | **84.28** |
| Qwen-VL | Regular | 83.37 | 81.57 | 84.53 | 82.58 | 85.23 | 83.09 | 84.37 | 82.41 |
| | VCD | 83.73 | 82.38 | 85.87 | 84.27 | 87.03 | 85.45 | 85.54 | 84.03 |
| | M3ID | 83.37 | 81.62 | 86.07 | 84.30 | 86.40 | 84.50 | 85.28 | 83.47 |
| | LightCD(ours) | **84.80** | **84.20** | **87.36** | **86.44** | **90.40** | **90.03** | **87.52** | **86.89** |

Table 1: **Main results on POPE_MSCOCO.** We evaluate the accuracy of various LVLMs on the POPE task. **Regular** denotes the setting using direct sampling. Values in **bold** indicate the highest performance.

**Results on POPE.** The experimental results of LightCD on the POPE are shown in Table 1(See the appendix A.1 for all results). As observed, when the negative samples shift from random to adversarial, the performance of LVLMs degrades significantly (Accuracy drops 4.33 and F1 score drops 3.76). This indicates that the prior knowledge embedded in LVLMs can interfere with model judgment, leading to hallucinations by ignoring the actual image content. Contrastive decoding methods are designed to mitigate hallucinations by counteracting such priors. For LLaVA, LightCD achieves average improvements of 6.11 in accuracy and 7.45 in F1 score. On InstructBLIP, the gains reach 6.46 and 5.83, respectively, while for Qwen-VL, LightCD improves performance by 3.15 and 4.48. Compared with other baselines, LightCD maintains a clear advantage even under the most challenging adversarial settings. Specifically, LightCD surpasses SID by 2.85 F1 points on LLaVA, outperforms SID by 1.20 points on InstructBLIP, and exceeds VCD by 1.82 points on Qwen-VL. Overall, LightCD consistently delivers state-of-the-art performance across different models and datasets, demonstrating its strong effectiveness in mitigating hallucinations.

**Results on CHAIR.** We evaluate generative object hallucination using CHAIR$_s$ and CHAIR$_I$, under maximum generation lengths of 64 and 128. As shown in Table 2, LightCD consistently outperforms

| Method | LLaVA-1.5 | | | | Qwen-VL | | | |
|---|---|---|---|---|---|---|---|---|
| | Max Token 64 | | Max Token 128 | | Max Token 64 | | Max Token 128 | |
| | CHAIR$_S$ ↓ | CHAIR$_I$ ↓ | CHAIR$_S$ ↓ | CHAIR$_I$ ↓ | CHAIR$_S$ ↓ | CHAIR$_I$ ↓ | CHAIR$_S$ ↓ | CHAIR$_I$ ↓ |
| Regular | 26.2 | 9.4 | 55.0 | 16.3 | 33.6 | 12.9 | 52.0 | 16.5 |
| VCD | 24.4 | 7.9 | 54.4 | 16.6 | 33.0 | 12.8 | 50.2 | 16.8 |
| M3ID | 21.4 | 6.3 | 56.6 | 15.7 | 32.2 | 11.5 | 49.5 | 17.2 |
| Woodpecker | 24.9 | 7.5 | 57.6 | 16.7 | 31.1 | 12.3 | 51.8 | 16.3 |
| HALC | 21.7 | 7.1 | 51.0 | 14.8 | 28.2 | 9.1 | 49.6 | 15.4 |
| ONLY | **20.0** | **6.2** | 49.8 | 14.3 | 27.3 | 8.4 | 48.0 | 14.3 |
| **LightCD** | 23.4 | 8.2 | **48.6** | **14.2** | **23.6** | **7.2** | **43.6** | **12.5** |

Table 2: **Results on CHAIR benchmark.** We limit the maximum number of new tokens to 64 or 128. Lower (↓) CHAIR$_S$, CHAIR$_I$ indicate better performance. The best results in each setting are **bolded**, and the second-best are underlined.

| Method | Object-level | | Attribute-level | | MME Score ↑ |
|---|---|---|---|---|---|
| | Existence ↑ | Count ↑ | Position ↑ | Color ↑ | |
| Regular | 173.75 | 121.67 | 117.92 | 149.17 | 562.50 |
| DoLa | 176.67 | 113.33 | 90.55 | 141.67 | 522.22 |
| OPERA | 183.33 | 137.22 | 122.78 | 155.00 | 598.33 |
| VCD | 186.67 | 125.56 | 128.89 | 139.45 | 580.56 |
| M3ID | 186.67 | 128.33 | 131.67 | 151.67 | 598.11 |
| SID | 187.50 | 125.00 | 126.66 | 149.17 | 588.33 |
| HALC | 183.33 | 133.33 | 107.92 | 155.00 | 579.58 |
| ONLY | 191.67 | **145.55** | **136.66** | 161.66 | 635.55 |
| **Ours** | **195** | 143.33 | 126.66 | **180** | **644.99** |

Table 3: **Results on MME-Hallucination (Fu et al., 2025) with LLaVA-1.5 (Liu et al., 2024b).** We report the average MME scores for each subset. Higher scores (↑) indicate better performance. The best results are **bolded**, and the second-best are underlined.

all baseline methods. The improvement is especially pronounced on Qwen-VL(max token =64), LightCD reduces hallucination rates by 10 points on CHAIR$_s$ and 5.7 points on CHAIR$_I$ compared with the regular. In LLaVA, a comparison between LightCD and the second-best method, ONLY, shows that LightCD's advantage grows as the generation length increases, leading to more effective hallucination mitigation. This indicates that other approaches struggle to handle hallucinations when generating long textual outputs, whereas LightCD—benefiting from its targeted design—can better suppress excessive language priors and thus produce outputs that are more objective and faithful to the visual content.

**Results on MME.** Table 3 presents a comparison between LightCD and other baselines on the MME dataset. LightCD achieves the best overall performance across both object-level hallucination (existence, count) and attribute-level hallucination (position, color). In particular, it shows substantial advantages on the existence and color sub-tasks. Although LightCD performs slightly less favorably on the counting and location tasks, it still delivers the strongest overall results. These findings underscore the effectiveness of our approach in mitigating hallucinations.

**Results on MM-Vet and MMBench.** The MMBench and MM-Vet benchmarks cover a broad range of tasks—including mathematics, OCR, and visual understanding—providing a more comprehensive evaluation. The corresponding results on LLaVA are presented in Table 4. LightCD achieves competitive results on MMBench, while on MM-Vet it significantly outperforms all other methods. Notably, in complex tasks such as math—where most baselines degrade model performance—LightCD improves over the second-best method, Opera, by 7 points. This suggests that LightCD remains highly effective even under challenging scenarios, substantially enhancing the model's reasoning and understanding abilities.

| Method | MMBench | MM-Vet | | | | | | |
|---|---|---|---|---|---|---|---|---|
| | Total | Rec | OCR | Know | Gen | Spat | Math | Total |
| Regular | 64.1 | 30.8 | 19.0 | 14.5 | 17.9 | 26.9 | 11.5 | 26.1 |
| VCD | 64.6 | 35.6 | 21.9 | 18.3 | 21.9 | 28.9 | 3.8 | 30.9 |
| M3ID | 64.4 | 35.0 | 19.7 | 18.8 | 19.0 | 26.0 | 7.7 | 29.9 |
| DoLA | 64.4 | 37.2 | 22.1 | 17.9 | 21.0 | 26.3 | 7.7 | 31.7 |
| OPERA | 64.0 | 35.4 | 25.6 | 20.5 | 22.9 | 30.9 | 11.5 | 32.0 |
| HALC | 64.2 | 36.2 | 21.5 | 17.5 | 20.1 | 23.5 | 7.7 | 30.8 |
| ONLY | **65.0** | 37.3 | 23.9 | **22.9** | 22.1 | 31.3 | 3.8 | 32.8 |
| **LightCD(ours)** | 64.6 | **38.1** | **30.4** | 21.2 | **24** | **36.1** | **18.5** | **35.2** |

Table 4: **Detailed results on MM-Vet and MMBench benchmark**. Abbreviations adopted: Rec for Recognition, OCR for Optical Character Recognition, Know for Knowledge, Gen for Language Generation, Spat for Spatial Awareness, Math for Mathematics. The best results are **bolded**, and the second best are underlined.

# 6 ANALYSIS AND DISCUSSIONS

In this section, we conduct ablation studies on inference latency, hyperparameter sensitivity, and a case study. Additional analyses, including the impact of SAP and Residual-free mechanism, and different sampling strategies, are provided in Appendix A.2.

| Method | First-Token↓ | 50-Token↓ | Memory↓ |
|---|---|---|---|
| LLaVA | 77.0 | 1484.0 | 15993 |
| ICD | 158.3↑ × 2.1 | 3128.3↑ × 2.1 | 16887 |
| VCD | 162.4↑ × 2.1 | 2778.3↑ × 1.9 | 16877 |
| SID | 164.4↑ × 2.1 | 3014.4↑ × 2.0 | 16961 |
| $SID_{FastV}$ | 153.4↑ × 2.0 | 2850.5↑ × 1.9 | 19283 |
| LightCD | 108.7↑ × 1.4 | 1529.1↑ × 1.0 | 16385 |

Table 5: **Inference latency** Comparison of the efficiency of various methods for generating tokens of different lengths on an NVIDIA A100 GPU.

Figure 4: **Hyperparameter Sensitivity of** $k$ **and** $\gamma$ on POPE.

**Inference Latency across Different Methods.** To improve the efficiency of contrastive decoding, LightCD optimizes the hallucination generation process, enabling a lightweight design that significantly outperforms existing methods in inference speed. Table 5 presents the results under the adversarial setting of the POPE-COCO dataset. We evaluate the inference latency of different methods from three aspects: first-token generation time, latency for generating a sentence of 50 tokens, and GPU memory usage.

In first-token generation stage, since the KV cache is not yet activated, the model must compute all key and value vectors from scratch. As a result, methods such as VCD, ICD, and SID exhibit roughly 2× the inference time compared to the original LLaVA. In contrast, LightCD only modifies the final layer, reducing computation and achieving a first-token latency of just 1.4× that of the base model. When generating an average-length sentence of 50 tokens, the inference time for VCD (Leng et al., 2024), ICD (Wang et al., 2024), and SID (Huo et al., 2025) increases to approximately 1.9×, 2.1×, and 2.0×, respectively. While SID with FastV (Chen et al., 2024b) improves speed compared to its standard mode, this comes at the cost of notably increased GPU memory consumption, which can be problematic in resource-constrained deployment scenarios.

Thanks to its lightweight design, LightCD achieves inference speed nearly identical to that of the original LLaVA, while also maintaining lower memory usage. These results demonstrate that LightCD effectively addresses the efficiency bottlenecks of contrastive decoding methods, delivering strong performance gains without sacrificing practicality.

**Hyperparameters Sensitivity Analysis.** Selective attention perturbation is a core component enabling LightCD to achieve lightweight contrastive decoding. We use two hyperparameters, $k$ and $\gamma$, to control the number of attention heads to be perturbed and the proportion of key tokens to be filtered, respectively. In this section, we analyze the sensitivity of LightCD to these hyperparameters.

We fix one parameter while varying the other, and evaluate model performance using accuracy and F1 score. The results are shown in Figure 4.

As observed, the performance of LightCD first improves and then declines as $k$ increases. A similar trend is seen for $\gamma$. This is because small values of $k$ and $\gamma$ fail to effectively suppress knowledge strongly aligned with the prompt, whereas overly large values may introduce noise, negatively impacting the quality of contrastive decoding. Empirically, we find that on the LLaVA model, LightCD achieves strong results when $k = 28$ and $\gamma = 0.3$, meaning the top 30% of salient tokens in the sequence are filtered.

**Case Study.** As illustrated in Figure 5, we conducted a case study comparing the original LLaVA,

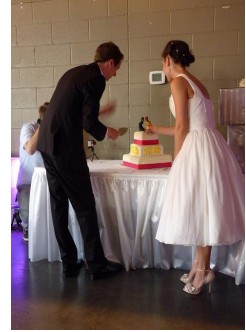

**Prompt:**
Please describe this picture in detail.

[LLaVA] The image captures a beautiful moment of a bride and groom cutting their wedding cake together. The bride is wearing a white dress, and the groom is dressed in a suit. They are standing close to each other, holding a knife together to cut the cake. The cake is placed on a dining table, which is surrounded by chairs. There are two chairs on the left side of the table, one on the right side, and another one at the far end. The scene appears to be set in a dining area, with a refrigerator visible in the background.

[VCD] The image captures a special moment at a wedding reception, where a bride and groom are standing together, cutting their wedding cake. The bride is wearing a beautiful white dress, and the groom is dressed in a suit. They are both holding a knife to cut the cake, which is placed on a dining table in front of them. There are several other people in the scene, some of them standing close to the couple, while others are further away. A camera is also visible in the background, likely capturing the memorable event. The atmosphere appears to be joyful and celebratory, as the couple shares this significant moment with their loved ones.

[SID] The image captures a beautiful bride and groom standing together in front of a dining table, cutting their wedding cake. The bride is wearing a white dress, and the groom is dressed in a suit. They are both holding a knife, preparing to cut the cake. The dining table is adorned with a cake placed in the center, surrounded by a few chairs. The couple appears to be the main focus of the scene, as they share this special moment together.

[LightCD] The image features a bride and groom standing next to each other at a wedding reception, cutting a large, multi-layered wedding cake together. The couple is dressed in formal attire, with the groom wearing a tie, and the bride in a beautiful white dress. The cake is placed on a dining table, which occupies a significant portion of the scene. The couple is holding a knife, ready to cut into the cake, while the guests, including another person visible in the background, look on, likely anticipating the moment. The atmosphere appears to be joyful, as the couple shares this special moment with their loved ones.

Figure 5: Comparison of different contrastive decoding methods, with hallucinated content highlighted in red.

VCD, SID, and LightCD. The LLaVA without contrastive decoding is highly susceptible to prior knowledge, often hallucinating nonexistent objects such as a chair or a refrigerator due to their frequent co-occurrence in similar scenes. Contrastive decoding partly alleviates this issue: SID removes the refrigerator but still misidentifies a chair, while VCD avoids nonexistent objects yet mistakenly detects multiple people. In contrast, LightCD yields more reliable and fine-grained outputs, correctly describing even a partially occluded person at the image boundary—"while the guests, including another person visible in the background." These findings show that LightCD suppresses spurious linguistic priors, thereby reducing hallucinations and producing more trustworthy and semantically accurate outputs. This case study underscores the advantages of LightCD in enhancing both the reliability and descriptive granularity of LVLMs.

## 7 CONCLUSIONS

In this work, we propose LightCD, a lightweight contrastive decoding approach designed to mitigate hallucinations in LVLMs while addressing the computational cost and inference latency limitations of prior methods. LightCD is built upon two key components: Selective Attention Perturbation, which identifies attention heads via an entropy-based criterion and filters their critical information, and a Residual-free Mechanism, which suppresses early-layer information to amplify controllable hallucinations. Experiments show that LightCD significantly outperforms existing methods in mitigating hallucinations while offering clear advantages in both inference latency and memory usage.

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

# A  APPENDIX

## A.1  ALL RESULTS ON POPE

Table 6 reports the results of LightCD on the full POPE dataset, which consists of three subsets: MSCOCO, A-OKVQA, and GQA. For each subset, negative samples are constructed using three strategies: random, popular, and adversarial. The results show that adversarial negatives (i.e., co-occurrence) pose a greater challenge to the model's comprehension ability, often leading to more severe hallucinations. In contrast, LightCD substantially improves the performance of LVLMs across all subsets, demonstrating its effectiveness in mitigating hallucinations. By better suppressing language priors, LightCD enables the model to generate more reliable responses.

## A.2  MORE ABLATION ANALYSIS

**The Effect of SAP and Residual-free mechanism**

LightCD leverages two core components—Selective Attention Perturbation (SAP) and the Residual-free Mechanism—to achieve lightweight contrastive decoding and effectively mitigate inference latency. We conduct ablation studies on both LLaVA and Qwen-VL, with results shown in Figure 6. In our ablation setup, Regular, only_SAP, only_Res, and LightCD denote the original model, the variant using only SAP, the variant using only the Residual-free mechanism, and the full LightCD configuration, respectively

Table 6: Results on the full POPE dataset with LLaVA-1.5 7B and InstructBLIP 7B.

| Dataset | Setting | Model | Decoding | Accuracy ↑ | Precision | Recall | F1 Score ↑ |
|---|---|---|---|---|---|---|---|
| COCO | Random | LLaVA-1.5 | Regular | 83.29 | 92.13 | 72.80 | 81.33 |
| | | | VCD | 87.00 | 86.13 | 88.20 | 87.15 |
| | | | M3ID | 87.50 | 87.38 | 87.67 | 87.52 |
| | | | LightCD | 89.90 | 91.53 | 87.93 | 89.69 |
| | | InstructBLIP | Regular | 80.71 | 81.67 | 79.19 | 80.41 |
| | | | VCD | 86.23 | 88.14 | 83.73 | 85.88 |
| | | | M3ID | 86.67 | 88.09 | 84.80 | 86.41 |
| | | | LightCD | 88.77 | 92.73 | 84.13 | 88.22 |
| | Popular | LLaVA-1.5 | Regular | 81.88 | 88.93 | 72.80 | 80.06 |
| | | | VCD | 83.10 | 79.96 | 88.33 | 83.94 |
| | | | M3ID | 84.30 | 81.58 | 88.60 | 84.95 |
| | | | LightCD | 87.97 | 91.30 | 83.93 | 87.46 |
| | | InstructBLIP | Regular | 78.22 | 77.87 | 78.85 | 78.36 |
| | | | VCD | 80.07 | 77.67 | 84.40 | 80.89 |
| | | | M3ID | 80.97 | 77.93 | 86.40 | 81.85 |
| | | | LightCD | 83.06 | 84.30 | 81.27 | 82.76 |
| | Adversarial | LLaVA-1.5 | Regular | 78.96 | 83.06 | 72.75 | 77.57 |
| | | | VCD | 77.17 | 72.18 | 88.40 | 79.47 |
| | | | M3ID | 78.23 | 73.51 | 88.27 | 80.22 |
| | | | LightCD | 84.57 | 85.20 | 83.67 | 84.43 |
| | | InstructBLIP | Regular | 75.84 | 74.30 | 79.03 | 76.59 |
| | | | VCD | 77.20 | 74.29 | 83.20 | 78.49 |
| | | | M3ID | 77.47 | 73.68 | 85.47 | 79.14 |
| | | | LightCD | 79.80 | 74.45 | 90.73 | 81.79 |
| VQA | Random | LLaVA-1.5 | Regular | 83.45 | 87.24 | 78.36 | 82.56 |
| | | | VCD | 83.83 | 78.05 | 94.13 | 85.34 |
| | | | M3ID | 84.67 | 79.25 | 93.93 | 85.97 |
| | | | LightCD | 88.67 | 84.69 | 94.4 | 89.28 |
| | | InstructBLIP | Regular | 80.91 | 77.97 | 86.16 | 81.86 |
| | | | VCD | 84.20 | 80.90 | 89.53 | 85.00 |
| | | | M3ID | 85.43 | 81.77 | 91.20 | 86.23 |
| | | | LightCD | 88.13 | 86.39 | 90.53 | 88.41 |
| | Popular | LLaVA-1.5 | Regular | 79.90 | 80.85 | 78.36 | 79.59 |
| | | | VCD | 76.63 | 69.59 | 94.60 | 80.19 |
| | | | M3ID | 77.80 | 70.98 | 94.07 | 80.91 |
| | | | LightCD | 82.37 | 76.12 | 94.33 | 84.25 |
| | | InstructBLIP | Regular | 76.19 | 72.16 | 85.28 | 78.17 |
| | | | VCD | 78.63 | 73.53 | 89.47 | 80.72 |
| | | | M3ID | 78.80 | 73.38 | 90.40 | 81.00 |
| | | | LightCD | 83.83 | 83.46 | 84.40 | 83.92 |
| | Adversarial | LLaVA-1.5 | Regular | 74.04 | 72.08 | 78.49 | 75.15 |
| | | | VCD | 67.40 | 61.39 | 93.80 | 74.21 |
| | | | M3ID | 68.60 | 62.22 | 94.73 | 75.11 |
| | | | LightCD | 75.37 | 68.73 | 93.07 | 79.07 |
| | | InstructBLIP | Regular | 70.71 | 65.91 | 85.83 | 75.56 |
| | | | VCD | 71.00 | 65.41 | 89.13 | 75.45 |
| | | | M3ID | 70.10 | 64.28 | 90.47 | 75.16 |
| | | | LightCD | 76.40 | 73.08 | 83.60 | 77.99 |
| GQA | Random | LLaVA-1.5 | Regular | 83.73 | 87.16 | 79.12 | 82.95 |
| | | | VCD | 83.23 | 76.73 | 95.40 | 85.05 |
| | | | M3ID | 84.20 | 78.00 | 95.27 | 85.77 |
| | | | LightCD | 86.70 | 81.77 | 94.47 | 87.66 |
| | | InstructBLIP | Regular | 79.65 | 77.14 | 84.29 | 80.56 |
| | | | VCD | 82.83 | 80.16 | 87.27 | 83.56 |
| | | | M3ID | 83.07 | 80.06 | 88.07 | 83.87 |
| | | | LightCD | 85.90 | 85.40 | 86.60 | 86.00 |
| | Popular | LLaVA-1.5 | Regular | 78.17 | 77.64 | 79.12 | 78.37 |
| | | | VCD | 72.37 | 65.27 | 95.60 | 77.58 |
| | | | M3ID | 73.87 | 66.70 | 95.33 | 78.49 |
| | | | LightCD | 83.60 | 79.24 | 91.07 | 84.74 |
| | | InstructBLIP | Regular | 73.87 | 69.63 | 84.69 | 76.42 |
| | | | VCD | 76.13 | 71.10 | 88.07 | 78.68 |
| | | | M3ID | 75.17 | 69.94 | 88.27 | 78.04 |
| | | | LightCD | 77.30 | 73.02 | 86.60 | 79.23 |
| | Adversarial | LLaVA-1.5 | Regular | 75.08 | 73.19 | 79.16 | 76.06 |
| | | | VCD | 68.83 | 62.26 | 95.67 | 75.43 |
| | | | M3ID | 68.67 | 62.16 | 95.40 | 75.28 |
| | | | LightCD | 78.40 | 72.71 | 90.93 | 80.81 |
| | | InstructBLIP | Regular | 70.56 | 66.12 | 84.33 | 74.12 |
| | | | VCD | 71.00 | 65.75 | 87.67 | 75.14 |
| | | | M3ID | 71.17 | 65.79 | 88.20 | 75.36 |
| | | | LightCD | 74.56 | 70.01 | 85.93 | 77.16 |

Table 7: Results on the full POPE dataset with QWen-VL.

| Dataset | Setting | Decoding | Accuracy ↑ | Precision | Recall | F1 Score ↑ |
|---------|---------|----------|-----------|-----------|--------|-----------|
| COCO | Random | Regular | 85.23 | 97.23 | 72.53 | 83.09 |
| | | VCD | 87.03 | 97.36 | 76.13 | 85.45 |
| | | M3ID | 86.40 | 98.23 | 74.13 | 84.50 |
| | | LightCD | 88.46 | 95.64 | 80.60 | 87.48 |
| | Popular | Regular | 84.53 | 94.50 | 73.33 | 82.58 |
| | | VCD | 85.87 | 94.98 | 75.73 | 84.27 |
| | | M3ID | 84.95 | 80.97 | 77.93 | 86.40 |
| | | LightCD | 87.36 | 93.21 | 80.60 | 86.45 |
| | Adversarial | Regular | 83.37 | 91.47 | 73.60 | 81.57 |
| | | VCD | 83.73 | 89.84 | 76.07 | 82.38 |
| | | M3ID | 83.37 | 91.19 | 73.87 | 81.62 |
| | | LightCD | 84.80 | 87.45 | 81.27 | 84.24 |
| VQA | Random | Regular | 86.40 | 94.32 | 77.47 | 85.07 |
| | | VCD | 87.93 | 94.59 | 80.47 | 86.96 |
| | | M3ID | 87.50 | 95.33 | 78.87 | 86.32 |
| | | LightCD | 89.43 | 91.74 | 86.67 | 89.13 |
| | Popular | Regular | 85.77 | 92.82 | 77.53 | 84.49 |
| | | VCD | 87.33 | 93.68 | 80.07 | 86.34 |
| | | M3ID | 87.37 | 95.31 | 78.60 | 86.15 |
| | | LightCD | 87.57 | 88.31 | 86.60 | 87.45 |
| | Adversarial | Regular | 80.37 | 82.56 | 77.00 | 79.68 |
| | | VCD | 81.90 | 83.07 | 80.13 | 81.57 |
| | | M3ID | 81.90 | 84.25 | 78.47 | 81.26 |
| | | LightCD | 80.60 | 77.32 | 86.6 | 81.70 |
| GQA | Random | Regular | 85.10 | 91.42 | 77.47 | 83.87 |
| | | VCD | 87.00 | 92.11 | 80.93 | 86.16 |
| | | M3ID | 87.07 | 92.64 | 80.53 | 86.16 |
| | | LightCD | 84.83 | 89.61 | 78.80 | 83.86 |
| | Popular | Regular | 74.93 | 69.33 | 89.40 | 78.10 |
| | | VCD | 82.53 | 83.52 | 81.07 | 82.27 |
| | | M3ID | 82.68 | 83.74 | 80.85 | 82.27 |
| | | LightCD | 84.87 | 89.56 | 78.93 | 83.91 |
| | Adversarial | Regular | 78.77 | 79.33 | 77.80 | 78.56 |
| | | VCD | 81.17 | 81.48 | 80.67 | 81.07 |
| | | M3ID | 81.90 | 83.07 | 80.13 | 81.57 |
| | | LightCD | 78.87 | 78.87 | 78.87 | 78.87 |

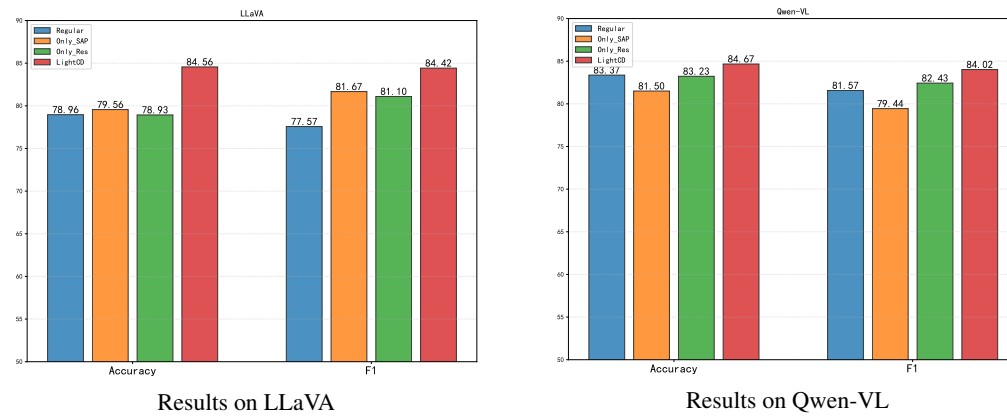



Results on LLaVA             Results on Qwen-VL

Figure 6: The effect of SAP and Residual-free mechanism on POPE


From the Figure 6, we observe that Only_SAP yields only marginal improvements over the original model on Accuracy. A reasonable hypothesis is that information propagated through residual connections from earlier layers counteracts the effect of SAP—namely, the key information suppressed by SAP is partially restored by the residual pathway, thereby diminishing the quality of the hallucination signal and ultimately affecting the final decoding performance. Similarly, Only_Res yields limited gains—although it blocks earlier-layer information, this disruption alone is insufficient to produce hallucinations of sufficient quality for effective contrastive decoding.

Overall, these results demonstrate that the combination of SAP and the residual-free mechanism jointly enables lightweight yet high-quality hallucination generation, which in turn allows contrastive decoding to effectively mitigate hallucinations in LVLMs and ensures the reliability of the final outputs.

**Different Decoding Stragey**

| Decoding | Method | Accuracy | Precision | Recall | F1 |
|----------|--------|----------|-----------|--------|-----|
| Top-P | LLaVA | 79.67 | 74.13 | 91.13 | 81.76 |
|        | LightCD | 84.60 | 85.60 | 83.20 | 84.38 |
| Top-K | LLaVA | 77.27 | 72.67 | 87.40 | 79.36 |
|       | LightCD | 83.77 | 84.10 | 83.27 | 83.69 |
| Temp. | LLaVA | 79.07 | 74.04 | 89.53 | 81.05 |
|       | LightCD | 82.77 | 80.06 | 87.27 | 83.51 |
| P + Temp. | LLaVA | 79.8 | 74.43 | 90.80 | 81.80 |
|           | LightCD | 82.6 | 79.93 | 87.07 | 83.34 |
| K + Temp. | LLaVA | 79.23 | 74.18 | 89.67 | 81.20 |
|           | LightCD | 82.13 | 79.53 | 86.53 | 82.89 |
| Greedy | LLaVA | 80.23 | 74.99 | 90.73 | 82.11 |
|        | LightCD | 82.97 | 80.24 | 87.47 | 83.70 |



Table 8: An ablation study of different decoding strategies on POPE.


To comprehensively evaluate the performance of LightCD under different decoding strategies, we extended our experiments beyond standard sampling and adopted six additional methods: Top-P sampling (p = 0.8), Top-K sampling (k = 50), Temperature sampling (t = 0.6), Top-P(p = 0.8) with

Temperature (t = 0.6), Top-K (k = 50) with Temperature (t = 0.6), and greedy decoding. The results are presented in Table 8.

As shown in the results, LightCD consistently demonstrates strong and stable performance across all decoding strategies, particularly in terms of accuracy and F1 score. Notably, the original LLaVA model achieves high recall but suffers from low precision. This suggests that the model's strong prior knowledge biases it toward giving affirmative responses, leading to frequent false positives by predicting the existence of objects that are not actually present.

LightCD addresses this issue by intervening in the model's decision token to generate contrastive hallucinations that help suppress the influence of such priors. As a result, it effectively mitigates hallucination-related errors and achieves significant improvements in both accuracy and F1 score across diverse decoding settings.

## A.3    AI USAGE INSTRUCTIONS

During the writing process of this paper, Large Language Models (LLMs) were only used to assist with language processing, including language translation, grammatical error checking, and sentence polishing. The key content of the paper, such as the core research ideas, framework design, reasoning process, and conclusions, was independently developed and completed by the author. LLMs were not involved in the generation of any content related to core originality or academic substance.

