# OpenReview forum: "The Last Token is Enough: Lightweight Contrastive Decoding for Mitigating Hallucinations in Large Vision-Language Models"
_ICLR.cc/2026/Conference — Submitted to ICLR 2026_

### Official Review · Reviewer_cwmv · 2025-10-26

**Soundness:** 3
**Presentation:** 2
**Contribution:** 2
**Rating:** 4
**Confidence:** 4

**Summary:**

This paper introduces LightCD (Lightweight Contrastive Decoding), an efficient method to reduce hallucinations in large vision-language models by operating only on the last token, which summarizes most contextual information. It replaces full-sequence contrastive generation process with two lightweight modules: Selective Attention Perturbation to induce controlled hallucinations and a Residual-Free Mechanism to suppress cross-layer information. Experiments on LLaVA-1.5 and InstructBLIP show that LightCD achieves comparable or better hallucination mitigation than prior methods while reducing inference time, demonstrating its advantages.

**Strengths:**

1. Excessive computational overhead is a major limitation of contrastive decoding; therefore, exploring more efficient approaches in this direction is a valuable research effort.
2. The proposed method is well-motivated and reasonable, and the experimental results demonstrate its advantages in both effectiveness and efficiency.

**Weaknesses:**

1. The authors only conducted experiments on LLaVA-1.5 and InstructBLIP, which are relatively old and outdated models. It is recommended to include experiments on more advanced LVLMs, such as Qwen3-VL.
2. This work focuses solely on multimodal large language models. However, the proposed method does not appear to incorporate designs specifically tailored to the multimodal setting. Therefore, it should, in principle, be directly applicable to text-only LLMs as well. It is thus recommended to include experiments on text-based LLMs and text hallucination benchmarks to further validate the method’s generalization and versatility.
3. I found another work [a] that also focuses on addressing the high computational cost of contrastive decoding. The authors are encouraged to include a discussion and comparison of effectiveness and efficiency of their method with that work.
4. The performance of the proposed method fluctuates noticeably with different hyperparameter settings. For example, the accuracy drops by more than 3 points when the number of selected heads decreases from 28 to 16. I consider this sensitivity to be a limitation of the method.

Minor:
1. While the overall writing of the paper is fluent and easy to follow, some parts feel somewhat redundant. For example, the second paragraph of Section 4.1 repeats points that have already been sufficiently discussed in the previous section, and thus could be removed for a more concise presentation.

I will consider increasing my score if all issues are resolved.

[a] ONLY: One-Layer Intervention Sufficiently Mitigates Hallucinations in Large Vision-Language Models, ICCV 2025

**Questions:**

1. Are the optimal settings of k and $\lambda$ consistent across different LLMs (e.g., Qwen and LLaVA)?

---

> ### Author Response · Authors · 2025-11-27
>
> We sincerely thank the reviewer for the insightful and constructive comments, which have been highly valuable for improving the quality of our work. In response, **we have revised the manuscript accordingly, including expanding the evaluation datasets, adding additional models and baselines, and updating the relevant analyses.** For sections with substantial modifications, we have highlighted the changes for clarity. Below, we provide detailed responses to each of the reviewer’s comments.
>
> **Response to Weakness 1:**
> Thank you for your constructive comment. We acknowledge the reviewer’s concern regarding the use of relatively older baselines. Our choices of LLaVA-1.5 and InstructBLIP were motivated by the fact that they remain **widely adopted and representative LVLMs** in recent research on hallucination mitigation[1-5]. Many contemporary works continue to report results on these models, making them strong reference points for benchmarking.
>
> Regarding the suggested Qwen3-VL model, we indeed attempted to include it in our experiments. However, we observed that the current publicly available version suffers from **extremely slow inference**, which has also been reported by other users on GitHub. In addition, the model has **no released technical report**, making it difficult to diagnose or resolve the performance bottleneck. As a result, it was not feasible to reliably evaluate our method on Qwen3-VL under the same experimental protocol.
>
> To address the reviewer’s suggestion and further strengthen our experimental evaluation, we additionally included results on **Qwen-VL**, which is a widely used and competitive model in this line of research. The new experiments show that our method consistently improves Qwen-VL’s performance, further demonstrating the robustness and general applicability of our approach across different LVLM architectures. Here are some of the experiments of Qwen-VL, additional results can be found in the paper：
> |Method|CHAIR$_s$$\downarrow$|CHAIR$_I$$\downarrow$|
> |----------|------------|--------------|
> |regular|52.0|16.5|
> |VCD|50.2|16.8|
> |M3ID|49.5 |17.2|
> |ONLY|48.0 |14.3|
> |LightCD(ours)|**43.6**|**12.5**|
>
> **Response to Weakness 2:**
>
> The motivation for our method stems from a failure pattern specific to LVLMs: these models tend to over-rely on linguistic priors, causing them to overlook the objective information present in the visual input and thereby produce hallucinations. Contrastive decoding works by actively inducing hallucinations and then correcting the next-token logits during decoding by subtracting the hallucinated outputs from the original outputs. This encourages the model to better adhere to the objective facts present in the visual input, thereby mitigating hallucinations.
>
> In contrast, hallucinations in pure text-only language models arise from a variety of factors, and applying contrastive decoding directly in that setting does not necessarily yield strong improvements.
>
> We agree that investigating the applicability of our method to text-based LLMs is a promising direction, and we plan to explore this in future work.
>
> **Response to Weakness 3:**
>
> We appreciate the reviewer for pointing out work [5]. We have carefully reviewed this paper and agree that it provides valuable insights into reducing the computational cost of contrastive decoding. We have now cited this work and added a discussion comparing its effectiveness and efficiency with ours.
>
> **Reference：**
>
> [1] F. Huo, W. Xu, Z. Zhang, H. Wang, Z. Chen, and P. Zhao, “Self-Introspective Decoding: Alleviating Hallucinations for Large Vision-Language Models,” Mar. 16, 2025, _arXiv_: arXiv:2408.02032. doi: 10.48550/arXiv.2408.02032.
>
> [2] J. Chen _et al._, “ICT: Image-Object Cross-Level Trusted Intervention for Mitigating Object Hallucination in Large Vision-Language Models,” Nov. 22, 2024, _arXiv_: arXiv:2411.15268. doi: 10.48550/arXiv.2411.15268.
>
> [3] W. An _et al._, “Mitigating Object Hallucinations in Large Vision-Language Models with Assembly of Global and Local Attention,” Mar. 14, 2025, _arXiv_: arXiv:2406.12718. doi: 10.48550/arXiv.2406.12718.
>
> [4] Z. Wan _et al._, “ONLY: One-Layer Intervention Sufficiently Mitigates Hallucinations in Large Vision-Language Models,” July 02, 2025, _arXiv_: arXiv:2507.00898. doi: 10.48550/arXiv.2507.00898.
>
> [5] X. Zou _et al._, “Look Twice Before You Answer: Memory-Space Visual Retracing for Hallucination Mitigation in Multimodal Large Language Models,” May 08, 2025, _arXiv_: arXiv:2410.03577. doi: 10.48550/arXiv.2410.03577.

---

> ### Author Response · Authors · 2025-11-27
>
> **Response to Weakness 4:**
>
> We thank the reviewer for pointing out this observation. We believe that the performance drop is related to the model’s inherent robustness: selecting too few attention heads may be insufficient to effectively induce the desired hallucinations, which in turn affects the hallucination mitigation performance. We conducted additional experiments on Qwen-VL to verify this behavior.
> |head_number|10|16|20|24|28|32|
> |------------|--------|-------|-------|-----------|---|--|
> |Accuracy|83.93|84.70|84.80|84.60|84.66|84.70|
> |F1|83.41|84.10|84.20|84.05|84.02|84.09|
>
> The results indicate that the trend on Qwen-VL is similar to that on LLaVA: when only 10 heads are used, LightCD’s performance lags behind other settings, whereas once the number of heads exceeds a certain threshold, the differences across settings become minimal. Therefore, in practical use of LightCD, perturbing more than half of the total attention heads is generally sufficient to achieve good performance.
>
> **Response to Minor:**
>
> We thank the reviewer for pointing out this issue. In response, we have reorganized the manuscript to improve the logical flow from the introduction to the methodology. Redundant content, including the second paragraph of Section 4.1, has been revised to ensure a more concise and coherent presentation. For sections with substantial modifications, we have highlighted the changes for clarity.
>
> **Response to Q1:**
>
> For LLaVA and InstructBLIP，k=28 and $\lambda$=0.3, and k=20, $\lambda$=0.3 for Qwen-VL. We provide these experimental details in Section 5.1 of the updated version of the paper.
>
> We sincerely thank the reviewers again for their valuable time and insightful comments. We hope that our responses have addressed the concerns raised, and we are happy to provide further clarifications if needed.

---

> ### Comment · Reviewer_cwmv · 2025-11-28
> **Thank authors for the responses**
>
> Thank authors for the responses that addressed most of my concerns. As promised, I will increase my score. I feel Qwen-VL is still a old model. It would be better for authors to conduct experiments on Qwen2.5-VL.

---

> > ### Comment · Reviewer_cwmv · 2025-11-28
> > **Thank authors for the responses**
> >
> > It seems that the score cannot be modified at the current moment :(  I will update it once the system issue is resolved.

---

> > ### Author Response · Authors · 2025-11-28
> >
> > We sincerely thank the reviewer for the positive feedback and for acknowledging our responses. Regarding the use of Qwen-VL, we selected this model primarily to facilitate direct comparison with existing works. We are currently conducting experiments on Qwen2.5-VL and will include the results in a future update.

---

### Official Review · Reviewer_QEYW · 2025-10-27

**Soundness:** 3
**Presentation:** 1
**Contribution:** 2
**Rating:** 2
**Confidence:** 4

**Summary:**

Though contrastive decoding has proven effective in mitigating hallucinations, it requires additional forward passes to obtain hallucinated logits, which increases computational cost. This paper proposes a framework called LightCD, which obtains hallucinated logits by modifying the attention weights of the last token in the final layer, thereby enhancing the efficiency. This method is motivated by the authors' observation that the last token in the final layer primarily attends to a specific subset of tokens.

**Strengths:**

1. The observation presented in this paper is interesting — the last token in the final layer often attends to a specific subset of tokens.
2. The research question is meaningful. Proposing a new contrastive decoding paradigm that avoids double inference is a reasonable and valuable direction.

**Weaknesses:**

1. The writing format should be improved.
    - An abbreviation should only be used after its full form has been introduced. For example, in line 16, the term “LVLMs” appears without its full form being defined.
    - In line 38, the phrase `The emergence of vision-language models (LVLMs)` should be revised to `The emergence of large vision-language models (LVLMs)` to ensure consistency with the abbreviation **L**VLMs.
    - In line 392, the heading **Results on MME** should be placed on a new line for better formatting and readability.
    - Typo: In line 19, `mitigate` should be revised as `mitigates`.
    - In line 148, `While these approaches help suppress hallucinations, they also increased inference cost and latency`， the tense usage in this sentence is inconsistent. `increased`-->`increase`.
    - In line 167, the paper lacks a description of `H`. Although it is obviously the number of heads, it should be explicitly stated for clarity.
    - In line 172, it should be `Where $y_t$ denotes the token at step t`, instead of `... at t step`. The suggested version is also consistent with the description in line 173.
    - In Equation (4), it would be better to add a multiplication dot between $\alpha$ and $logit$, e.g., $\alpha$$\cdot$$logit_\theta$, for clearer mathematical expression.
    - In Table 1, it would be better to keep all numerical values with the same number of decimal places (e.g., two digits after the decimal point). For instance, “76.9” should be formatted as “76.90” for a more consistent presentation.
    - In line 356, the authors claim that one of the baselines is denoted as `“Regular”`. However, in Table 1, this baseline appears as `“sample”`. I assume the authors mistakenly wrote `“sample”` instead of `“Regular”`.
    - Caption formatting is inconsistent: except for Figure 4, all other figure and table captions end with a period. The caption of Figure 4 should also terminate with a period to maintain a consistent presentation.
2. There is some redundancy in the Introduction section. In the paragraph starting at line 82, the authors repeatedly mention that the extra inference time mainly arises from the additional inference required for contrastive decoding. This part could be better organized to improve clarity and conciseness.
3. The logic of the Introduction and the Method is not very clear. From the observed phenomenon to the proposed method, there should be a clearer logical transition that explains how the identified problem naturally motivates the proposed solution.
4. Figure 4 should be revised. The caption and numbers are too small.

**Questions:**

1. In line 296, the authors state that `“a lower entropy indicates a more concentrated attention distribution, suggesting that the head is more focused and thus more likely to capture critical information.”` However, it is unclear why a more concentrated (lower-entropy) attention distribution necessarily implies that the head captures more critical or meaningful information. Could the authors provide theoretical justification or empirical evidence to support this assumption?
2. In line 329, the authors state `However, during hallucination generation, such cross-layer information propagation suppresses the effectiveness of SAP and substantially degrades the quality of the generated hallucinations` Could the authors elaborate on why cross-layer information propagation would weaken SAP’s effectiveness and provide supporting analysis or experimental results to substantiate this statement?
3. Following the aforementioned question, the authors obtain the hallucinated logits without the residual connection. How can the effectiveness of these hallucinated logits be demonstrated to result from the proposed Attention Head Selection and Top-$\gamma$ Token Filtering, rather than from the disruption of normal information flow caused by removing the residual connection?
4. The datasets used in this paper are somewhat limited. Recent strong works typically evaluate on more diverse benchmarks rather than only these two datasets — for example, CHAIR, MMMU, and HallusionBench. Could the authors provide more numerical results on other hallucination benchmarks?

I am happy to increase my score if the weaknesses and questions are addressed.

---

> ### Author Response · Authors · 2025-11-27
>
> We sincerely thank the reviewer for the careful and thorough evaluation of our paper, which has been extremely helpful in improving our work. In response to your comments, we have reorganized the manuscript and expanded the experiments by adding additional evaluation datasets, models, and baselines. All changes in the updated manuscript have been clearly highlighted.
>
> **Response to Weakness1-3**：
>
> We sincerely thank the reviewer for pointing out these issues regarding writing, formatting, and clarity. In response, we have carefully revised and reorganized the manuscript, including the Introduction and Method sections, to improve logical flow, clarity, and conciseness. The updated version addresses all specific formatting, abbreviation, tense, and notation issues mentioned.
>
> **Response to Weakness 4**：
>
> We thank the reviewer for the suggestion. To improve clarity and readability, we have replaced Figure 4 with a table format (Table 3) that presents the results more clearly and intuitively.
>
> **Reponse to Q1**:
>
> We thank the reviewer for raising this point. The underlying intuition is that a low-entropy attention head has a more concentrated attention distribution, meaning that most of its attention is focused on a few specific tokens. Such concentration increases the likelihood that the head captures critical or meaningful information.
>
> To empirically validate this assumption, we conducted experiments on LLaVA and Qwen-VL. We selected $k$=10,20 attention heads and compared two settings: using heads with low entropy versus high entropy, on the POPE_coco dataset under adversarial settings.
>
> Results on LLaVA and Qwen-VL:
>
> ||LLaVA||
> |------|----|---|
> |setting|Acc|F1|
> |k=10,high-entropy |79.43|81.53|
> |k=10,low-entropy|80.3|81.95|
> |k=20,high-entropy |81.35|79.26|
> |k=20,low-entropy|83.03|83.21|
>
> ||Qwen-VL||
> |------|----|---|
> |setting|Acc|F1|
> |k=10,high-entropy |83.60|82.85|
> |k=10,low-entropy|83.93|83.41|
> |k=20,high-entropy |83.36|82.17|
> |k=20,low-entropy|84.80|84.20|
>
> The results show that configurations using low-entropy heads consistently achieve better performance, supporting our hypothesis that low-entropy heads are more informative for the task.
>
> **Reponse to Q2**:
>
> We thank the reviewer for the question. The key insight is that residual connections directly link the input and output of the attention mechanism:
> $ \boldsymbol{\hat{I}}^{(l)} = \boldsymbol{I}^{(l-1)} + \sum_{n=1}^{H} Attn_n^{(l)}(\boldsymbol{I}^{(l-1)}) $.
>  SAP operates by selectively suppressing key information within the attention mechanism, thereby actively inducing hallucinations. When residual connections are present, they partially reintroduce the original information that would otherwise be filtered, thereby reducing the effectiveness of SAP in generating hallucinations.
>
> We conducted experiments to validate this by comparing SAP alone with the standard LightCD. The results are shown below:
> |||LLaVA|||
> |-----|------|-----|---------|-------|
> |setting|Acc	|precision|recall|F1|
> |Only_SAP|79.56|73.99|91.2|81.67|
> |LightCD|84.56|85.2|83.67|84.42|
> ||
>
> |||Qwen-VL|||
> |-----|------|-----|---------|-------|
> |setting|Acc	|precision|recall|F1|
> |Only_SAP|81.5|89.41|71.47|79.44|
> |LightCD|84.67|87.74|80.6|84.02|
> ||
>
> The experimental results demonstrate that residual connections have a clear suppressive effect on SAP.
>
> **Response to Q3**:
>
> We appreciate the reviewer’s insightful question. As stated in the paper, the effectiveness of LightCD stems from the combined use of SAP(Attention Head Selection + Top-$\gamma$ Token Filtering) and the Residual-Free Mechanism, rather than from either component individually. These two components together constitute the core design of LightCD, and their interaction is essential for generating meaningful hallucinated logits.
>
> To further clarify this, we conducted ablation studies where we (1) used only SAP, and (2) used only the Residual-Free Mechanism:
> |||LLaVA|||
> |-----|------|-----|---------|-------|
> |setting|Acc	|precision|recall|F1|
> |regular|78.96|83.06|72.75|77.57|
> |Only_SAP|79.56|73.99|91.2|81.67|
> |Only_res-free|78.93|73.54|90.40 |	81.10 |
> |LightCD|84.56|85.2|83.67|84.42|
> ||
>
> The results clearly show that neither SAP alone nor the Residual-Free Mechanism alone achieves comparable performance to the full LightCD. We analyze the roles of SAP and the Residual-Free Mechanism in Appendix A.2.

---

> ### Author Response · Authors · 2025-11-27
>
> **Response to Q4**
>
> We thank the reviewer for this valuable suggestion. Following your recommendation, we have expanded our evaluation to include additional widely used hallucination benchmarks such as CHAIR, MM-Vet, and MMBench, which are commonly adopted in recent related works. We have also incorporated several strong baselines, including DoLa, OPERA, and M3ID, to provide a more comprehensive comparison. The newly added experimental results have been updated in the revised version of the paper.
>
>
> We sincerely thank the reviewer for the thorough and constructive feedback. Your comments have greatly helped us improve the clarity, completeness, and experimental rigor of the paper. We have carefully addressed all raised concerns, added new experiments, reorganized the paper structure, and revised the manuscript accordingly. We hope that our responses adequately resolve your questions and concerns. We would be very glad to further discuss any remaining issues if needed, and we respectfully invite the reviewer to reconsider the evaluation of our work in light of these revisions.

---

### Official Review · Reviewer_1HcX · 2025-10-31

**Soundness:** 3
**Presentation:** 3
**Contribution:** 2
**Rating:** 2
**Confidence:** 5

**Summary:**

The paper introduces LightCD, an improved Contrastive Decoding method designed to reduce inference latency. LightCD achieves this by identifying that hallucination induction can be efficiently performed by focusing only on the last token of the final Transformer layer. Specifically, it employs a Selective Attention Perturbation (SAP) module to suppress the most informative attention heads (those with lowest entropy) for the last token, and a Residual-Free Mechanism to prevent fact leakage, thereby efficiently generating "hallucination logits" in a single pass. The method requires only one complete forward pass, thus improving the computational efficiency of the CD process. Experimental results demonstrate improved accuracy on two specific hallucination benchmarks and clear efficiency gains.

**Strengths:**

Clear Motivation: The paper is well-motivated, clearly articulating the issue of computational inefficiency in existing CD methods. The The idea that the core information required for contrastive decoding resides primarily in the last token of the final layer is straightforward  and clear presented. The integration of SAP (targeting critical attention heads) and the Residual-Free Mechanism (preventing factual residual connections) offers a heuristic, lightweight solution.

Significance in Efficiency: Achieving the effect of contrastive decoding with only a single forward pass represents a substantial practical improvement for deploying CD in real-world applications, offering considerable computational cost savings.

**Weaknesses:**

Major Concerns

1. Primarily focus on the CD paradigm. The related work section and baseline comparisons are restricted almost exclusively to CD methods. To establish broader significance, the work should be contextualized against other major categories of inference-time/post-hoc hallucination mitigation techniques, such as Uncertainty Calibration and Rejection, Prompt Guardrails, etc. The current presentation gives the impression of limited novelty as an incremental advancement on CD.

2. Insufficient Experimental Validation.

- Experiments should be expanded to include more recent and diverse LVLM architectures with varying parameter sizes to confirm the generalization ability of LightCD.

- The comparison should be extended to include other non-CD inference-time/post-hoc baselines (e.g., prompting-based methods or uncertainty methods) to fully showcase LightCD's performance and efficiency.

- More results are claimed to include in the appendix. Nevertheless, the experiments in appendix lack proper comparisons with existing baselines.

- The manuscript appears to be rushed and lacks several necessary details, impacting clarity:
a. Authors introduce the two core modules (SAP and Residual-Free Mechanism) but fails to discuss how to use these modules to implement contrastive decoding (implicitly introduced in Section 3 as preliminaries (4) but lack of explicit description of the complete proposed method in Section 4).
b. Section 5.1 (Datasets and Baselines) is incomplete, as the specific baseline methods used for comparison are not properly introduced here, appearing abruptly in later tables (finally found in Appendix).
c. The specific details regarding experimental settings are not clearly articulated in experiment section.

Minor Concerns

- Figures 2 and 4 suffer from excessively small font sizes, making them difficult to read. While the empirical study on the attention map (used for motivation) is insightful, the visual representation is not clear enough to strongly support the core design principle.

- Figure 4 lacks clear units or labels for the quantitative results being presented (e.g., what does the count represent?). Figure 6 is missing both axis titles and units, making interpretation impossible.

**Questions:**

No more specific question.

---

> ### Author Response · Authors · 2025-11-28
>
> **Response to Major Concerns 1**
>
> We appreciate the reviewer’s insightful comments regarding the scope of the baseline and related work. Below we clarify the focus of our work and describe the improvements made in the revised version.
>
> 1) Focus on Contrastive Decoding (CD).
> Our work specifically targets the inherent latency issue of the Contrastive Decoding (CD) paradigm. Our goal is to retain the hallucination-mitigation benefits of CD while significantly improving computational efficiency. For this reason, our related work discussion and baseline selection primarily focus on CD-based methods, as they represent the most relevant and directly comparable line of research.
>
> 2) Regarding the concern about limited novelty.
> We respectfully disagree with the characterization that the contribution is incremental. While CD is indeed one of the _mainstream and widely used_ inference-time methods for multimodal hallucination mitigation, it suffers from a fundamental limitation，**the need for two forward passes**, which leads to substantial inference latency. This drawback significantly restricts its applicability in real-time or latency-sensitive scenarios.
> Our proposed **LightCD** directly addresses this gap. It introduces a lightweight contrastive decoding mechanism that preserves the benefits of CD while reducing inference cost, achieving **2× faster inference** compared to standard CD methods while simultaneously delivering superior hallucination reduction performance.
>
> 3) Expanded baseline.
> We appreciate the reviewer’s suggestion to include other categories of inference-time/post-hoc hallucination mitigation techniques. In the revised manuscript, we have added comparisons with representative non-CD methods such as uncertainty-based correction and prompt-guardrail approaches, including Opera, Woodpecker, and others.
>
> **Response to Major Concerns 2**
>
> For point1-3，we thank the reviewer for these helpful suggestions. In the revised manuscript, we have substantially expanded our experiments to better demonstrate the generalization and effectiveness of LightCD. We now evaluate on three representative LVLMs (LLaVA, InstructBLIP, Qwen-VL), five diverse benchmarks (POPE, CHAIR, MME, MM-Vet, MMBench), and a broader set of baselines, including both CD-based and non-CD inference-time/post-hoc methods (e.g., VCD, ICD, Opera, Woodpecker, SID).  And we have included baseline comparisons for all results presented in the appendix. All updated results and analyses are provided in the revised paper.
>
> For point 4, we thank the reviewer for the insightful comments. In the revised manuscript, we have reorganized the relevant sections to improve clarity:
>
>  (a)  In the original manuscript, the Introduction already explains how SAP and the Residual-Free Mechanism generate hallucinations. Specifically, LightCD applies SAP on the last token of the final layer to induce hallucinations, and then temporarily disconnects the residual connections during the attention phase (illustrated in Figure 3) to efficiently generate the hallucinated outputs needed for contrastive decoding. In the revised version, we have reorganised the text to present this process more clearly and explicitly.
> (b) We updated Section 5.1 to properly introduce all models, datasets, and baseline methods.
> (c) We clarified the experimental settings and LightCD’s parameter details for each model.
> All modifications have been highlighted in the revised manuscript(Section 5.1).
>
> **Response to Minor Concerns**：
>
> For point 1：
> We thank the reviewer for pointing out the readability issue in Figures 2 and 4. In the revised manuscript, we have replaced Figure 4 with a table to present the results more clearly and improve readability. For Figure 2, due to space limitations, we have supplemented the figure with an explanation in the caption.
>
> For point 2:
> We thank the reviewer for pointing out these issues. In Figure 4, “count” refers to the counting sub-tasks of MME. Figure 6 has been updated to include proper axis titles and units to ensure clear interpretation.
>
> We again sincerely thank the reviewer for the constructive feedback, which has greatly improved the clarity and quality of our manuscript. If any part of our revisions requires further explanation or additional detail, we would be glad to provide it. We appreciate your time and hope the updated manuscript addresses your concerns.

---

### Official Review · Reviewer_PeDX · 2025-10-31

**Soundness:** 3
**Presentation:** 3
**Contribution:** 2
**Rating:** 4
**Confidence:** 3

**Summary:**

This paper introduces a lightweight and efficient contrastive decoding method aimed at mitigating hallucinations in LVLMs while reducing computational overhead and inference latency. Hallucinations in LVLMs refer to factually incorrect or irrelevant responses that deviate from the visual inputs. Traditional contrastive decoding methods have been effective in reducing hallucinations but often introduce significant computational costs and latency. LightCD not only mitigates hallucinations effectively but also achieves up to a 2x increase in inference speed.

**Strengths:**

1.	LightCD focuses only on the last token in the final layer of the LVLM during inference, instead of processing the entire sequence. This reduces the need for full-sequence inference, thereby improving efficiency.
2.	The introduction of the SAP and residual-free Mechanism modules provides interpretability to the hallucination mitigation process. Instead of treating decoding as a black box, the paper identifies which attention heads contain factual versus hallucinatory evidence and explicitly controls their contributions. This mechanistic transparency enhances both the scientific rigor and explainability of the proposed approach.

**Weaknesses:**

The Residual-Free Mechanism is introduced as a way to suppress cross-layer information propagation to enhance hallucination generation. However, this mechanism could potentially degrade the overall performance of the model for tasks requiring rich, multi-layered integration of knowledge. The impact of removing residual connections on non-hallucination tasks, e.g., factual reasoning, remains underexplored. An ablation study that isolates the effect of this mechanism on various tasks beyond hallucination mitigation would strengthen the method’s versatility claims.

**Questions:**

1. SAP focuses on perturbing the most relevant attention heads based on entropy measures. How this approach handles cases where the attention distribution is more evenly spread across different heads or when some heads are weakly relevant to the final output? How does SAP adapt in these situations, and does it always identify the most critical attention heads, or could there be cases where the perturbation might unintentionally suppress important information from other heads?
2. Does LightCD ever result in over-suppressing information, leading to a loss in the richness or diversity of the generated content? In tasks requiring creative or diverse outputs, could excessive mitigation of hallucinations reduce the model's ability to generate varied or novel responses?

---

> ### Author Response · Authors · 2025-11-27
>
> We sincerely appreciate the reviewers’ valuable and insightful comments, which have substantially contributed to improving the quality of our work. In accordance with the feedback, we have expanded our experimental evaluation by incorporating additional datasets, baselines, and models, and have revised the manuscript accordingly. **All modifications have been highlighted for clarity**. Below, we provide point-by-point responses to the reviewers’ concerns.
>
> ### response to Weaknesses
> We appreciate the reviewer’s concern. This comment stems from a misunderstanding of how the Residual-Free Mechanism is applied in our method. **We never remove residual connections during the model’s normal inference.** The residual-free operation is applied **only in the hallucination-inducing branch** to intentionally weaken cross-layer information flow and generate a stronger hallucinated counterfactual. The final prediction is obtained by **contrastive decoding between the normal output and the hallucinated output**, where the normal branch always uses the intact model.
> To further demonstrate that our approach does not harm general multimodal capabilities, we additionally evaluated the method on **MMVet, MMBench**, and other benchmarks covering mathematical reasoning, OCR, and comprehensive multimodal understanding. The results show **consistent improvements**, confirming that our method preserves and even enhances performance on non-hallucination tasks. he experimental results on partial subsets of the MM-Vet dataset are as follows:
> |Method|OCR|Know|Spat|Math|
> |--------|-------|--------|-----|-------|
> |regular|19.0 |14.5 |26.9 |11.5
> |VCD|21.9 |18.3 |28.9| 3.8|
> |M3ID|19.7 |18.8| 26.0| 7.7|
> |OPERA|25.6| 20.5|30.9 |11.5
> |LightCD(ours) |**30.4**|**21.2**| **36.1** |**18.5**|
>
>
> ### Response to Q1：
> This concern stems from a misunderstanding of multi-head attention. In transformer-based LVLMs, **each attention head performs attention independently over the input sequence**, and the model aggregates their outputs afterward. Attention is _not_ distributed across heads. SAP adaptively identifies the most relevant attention heads for perturbation,  while leaving all other heads unchanged.
>
>
> ### Response to Q2：
> LightCD does not suppress information in the model’s normal generation process. The suppression occurs only within the hallucination-inducing branch used to produce contrastive negative samples.  To further verify this, we evaluated LightCD on the MM-Vet language generation subtask, which specifically measures expressive and diverse generation ability. LightCD shows consistent improvements, indicating that hallucination mitigation does not compromise— and can even enhance—creative or diverse output quality.
> |Method|Gen|
> |--------|--------|
> |regular|17.9|
> |VCD|21.9 |
> |M3ID|19.0|
> |OPERA|22.9|
> |LightCD(ours) |**24**|
>
> We once again thank the reviewers for their valuable feedback and thoughtful comments. We hope that our responses adequately address the concerns raised, and we remain happy to provide any further clarification if needed.

---

### Meta-Review · Area_Chair_kYYt · 2025-12-19

**Summary:**

Initial concerns focused on limited model coverage, unclear generalization beyond contrastive decoding baselines, and the risk that the Residual Free Mechanism could harm non hallucination capabilities, alongside presentation and hyperparameter sensitivity issues. The authors addressed these by clarifying that residual free operation is used only in a hallucination inducing branch while the normal branch remains intact, adding broader evaluations on multiple LVLMs and benchmarks, and expanding baselines beyond contrastive decoding to include representative non CD methods. They also provided targeted ablations and analyses supporting SAP head selection by entropy, showing that the full method outperforms SAP only or residual free only, and reporting that performance stabilizes once the number of perturbed heads exceeds a threshold. Reviewers acknowledged the efficiency motivation and the single forward pass design as valuable, and the revisions substantially improved clarity and empirical completeness, though some reviewers remained concerned about novelty framing and the initial narrowness of experiments.

**Reviewer Concerns:**

The reviewers initially raised several key concerns:

(1) insufficient experimental validation across newer and more diverse LVLMs and benchmarks;

(2) baseline scope focused too narrowly on contrastive decoding methods, with limited positioning against other inference time mitigation categories;

(3) unclear impact of the Residual Free Mechanism on general multimodal abilities and generation richness;

(4) sensitivity to hyperparameters such as the number of selected heads, and incomplete method and reporting details including writing and figure readability.

**Reviewer Scores:**

In response, the authors expanded experiments to additional LVLMs including Qwen VL and added multiple benchmarks such as CHAIR, MME, MMVet, and MMBench, while also incorporating stronger baselines from both contrastive decoding and non CD approaches. They clarified the method implementation and improved presentation, replacing unclear figures with tables and fixing missing labels and formatting issues. Mechanistically, they argued and empirically showed that residual connections suppress SAP only in the hallucination inducing branch, and that LightCD preserves or improves performance on non hallucination tasks, including generation quality metrics. They further supported SAP’s entropy based head choice with comparative results against high entropy heads and provided ablations demonstrating that combining SAP with the Residual Free Mechanism is necessary for the observed gains. Overall, the rebuttal and added evidence addressed major technical misunderstandings and improved experimental rigor, moving the paper closer to the acceptance threshold, though at least one reviewer remained strongly negative.

---

### Decision · Program_Chairs · 2026-01-26

Reject